# The Carrier Conundrum; A Review of Recent Advances and Persistent Gaps Regarding the Carrier State of Foot-and-Mouth Disease Virus

**DOI:** 10.3390/pathogens9030167

**Published:** 2020-02-28

**Authors:** Carolina Stenfeldt, Jonathan Arzt

**Affiliations:** 1Foreign Animal Disease Research Unit, Agricultural Research Service, US Department of Agriculture, Plum Island animal Disease Center, Orient, NY 11957, USA; 2Department of Diagnostic Medicine/Pathobiology, Kansas State University, Manhattan, KS 66506, USA

**Keywords:** foot-and-mouth disease, foot-and-mouth disease virus, FMD, FMDV, virus, carrier, persistent infection, transmission, pathogenesis, cattle

## Abstract

The existence of a prolonged, subclinical phase of foot-and-mouth disease virus (FMDV) infection in cattle was first recognized in the 1950s. Since then, the FMDV carrier state has been a subject of controversy amongst scientists and policymakers. A fundamental conundrum remains in the discordance between the detection of infectious FMDV in carriers and the apparent lack of contagiousness to in-contact animals. Although substantial progress has been made in elucidating the causal mechanisms of persistent FMDV infection, there are still critical knowledge gaps that need to be addressed in order to elucidate, predict, prevent, and model the risks associated with the carrier state. This is further complicated by the occurrence of a distinct form of neoteric subclinical infection, which is indistinguishable from the carrier state in field scenarios, but may have substantially different epidemiological properties. This review summarizes the current state of knowledge of the FMDV carrier state and identifies specific areas of research in need of further attention. Findings from experimental investigations of FMDV pathogenesis are discussed in relation to experience gained from field studies of foot-and-mouth disease.

## 1. Introduction

Foot-and-mouth disease (FMD) is a viral disease of cloven-hoofed animals that has substantial impact on global agricultural production and trade [1]. Apart from causing a debilitating clinical condition which directly impacts animal production and welfare, the uneven global distribution of the disease creates substantial trade barriers that prevent access to international markets for many low- and middle-income countries due to the endemic presence of FMD virus (FMDV) [2]. FMD endemicity typically includes a cycle of periodic outbreaks of clinical FMD combined with both neoteric (temporally acute) and persistent phases of subclinical infection. Many FMD-endemic countries participate in the World Organization for Animal Health (OIE)-endorsed Progressive Control Pathway for FMD (PCP-FMD), which facilitates the stepwise process towards FMD control and ultimately eradication [3]. However, countries in which adequate FMD control has not been achieved suffer both from the direct impacts of the disease, including impaired food security and the costs of preventive and zoosanitary measures, as well as from loss of revenue due to imposed trade restrictions [2,4]. In contrast, countries that are currently free of FMD invest substantial resources into preparedness and surveillance of the global FMD situation, as a disease incursion would have catastrophic consequences for agricultural industries and far-reaching economic impacts. 

Foot-and-mouth disease virus is the prototype *Aphthovirus* within the *Picornaviridae* family. The virus is highly diverse, with seven immunologically distinct serotypes (O, A, C, Asia-1, and Southern African Territories [SAT] -1, -2, and -3), and multiple lineages and subtypes within serotypes [5,6]. The virus is capable of infecting a large range of both wild and domestic host species [7,8]. Although there are overarching similarities in clinical signs across animal species, the severity of clinical FMD varies greatly, depending upon the intrinsic characteristics of the virus strain and the susceptibility of the host. The phenomenon of subclinical FMDV infection in ruminants adds additional layers of complexity to FMD control, as the virus may be cryptically present in individual animals and populations, with variable and debatable relevance to epidemiology, transmission, and emergence of novel strains. Subclinical FMDV infection can be divided into two distinct phases: 1; neoteric subclinical infection which refers to acute-phase infection of vaccinated hosts or animals which are naturally resistant to the clinical disease and 2; persistent infection, also referred to as the FMDV carrier state. 

## 2. Overview of FMDV Pathogenesis

### 2.1. FMDV Infection and Definitions of Disease Stages

FMDV is infectious at very low doses, which vary depending on host species and route of exposure [9]. The severity of clinical FMD is co-determined by factors related to the host and the intrinsic variability of virulence across virus strains. This variation in virulence seems to be associated with distinct virus strains rather than serotypes. As an example of this, the naturally occurring FMDV O/Taiwan/1997 belonging to the Cathay topotype, causes severe and fulminant FMD in pigs [10,11], whereas cattle are only subclinically infected [12,13]. By contrast, essentially all other known serotype O FMDVs are equally virulent in cattle and pigs. 

Transmission of FMDV may occur via both direct and indirect contact between animals, as well as through fomites, and airborne spread under certain atmospheric conditions [14]. Fully susceptible animals will progress through clinically distinct stages of disease, whereas clinically protected animals progress through similar phases of infection with no clinically observable manifestation. This immunological protection may be vaccine-mediated or through natural resistance, as occurs with sympatric virus-host co-evolution.

Specifically, FMDV infection of non-immune hosts involves an initial incubation phase of subclinical (pre-clinical) infection [15,16] followed by systemic dissemination of the virus (viremia) and the onset of clinical disease, which defines the end of the incubation phase [17,18] (Figure 1).

The classical clinical phase of FMD is generally associated with varying degrees of vesicle-formation, lameness and inappetence, and may affect growth and milk yield as well as draught capacity. Similar to vaccinated cattle, subclinical FMDV infection is commonly reported in African buffalo (Syncerus caffer) [19]. The clinical manifestation of FMD is more variable but may be blunted in Asian buffalo (Bubalus bubalis) as well as in some sympatric cattle breeds in FMD-endemic regions [20,21,22]. Clinical FMD is often reported to be mild in small ruminants [23,24]. However, experimental studies have demonstrated that certain FMDV strains may cause severe clinical disease in sheep [25,26]. Additionally, similar to other ruminants, sheep may become persistently infected FMDV carriers, regardless of the occurrence of clinical disease [27,28,29].

Pigs are highly susceptible to FMDV infection, and the clinical phase of infection is often severe [30,31,32,33]. However, in contrast to ruminants, pigs efficiently clear infectious virus within four weeks of infection, and are not capable of maintaining persistent FMDV infection [34].

FMD-associated mortality is generally low, although death due to acute myocarditis occurs sporadically in juvenile animals and rarely in adults [33,35,36,37].

After the clinical phase, carrier-susceptible species enter a transitional phase wherein the virus is either cleared, or the animals transition to a persistent phase of infection. 

The term neoteric subclinical infection has been suggested to differentiate the early, or acute, stages of FMDV infection in clinically protected hosts from the subsequent persistent phase of infection [38]. Neoteric infection may refer to vaccinated animals or sympatric hosts which have co-evolved with endemic viruses. The most critical difference between neoteric and persistent subclinical FMDV infection relates to the substantially greater quantities of virus that are shed in oral- and nasal secretions during neoteric infection compared to the carrier state, through which infectious virus can generally only be recovered by the sampling of oropharyngeal fluid (OPF) using a probang cup [39,40,41]. 

### 2.2. Temporo-Anatomical Progression of FMDV Infection

Primary FMDV infection occurs within epithelial cells of the mucosa of the oropharynx or nasopharynx, depending on the host species. Cattle are highly susceptible to FMDV infection via inhalation, and the site of primary infection has been localized to specific regions of the nasopharyngeal epithelium that overlie mucosa-associated lymphoid tissue (MALT) [42,43,44,45,46]. Pigs are more effectively infected through oral exposure, with primary infection occurring within epithelial crypts of the oropharyngeal- and laryngopharyngeal tonsils [32,47]. Fewer efforts have been invested into detailed study of the early stages of FMDV infection of small ruminants, but current evidence suggests that primary infection in sheep occurs in either epithelial crypts of the oropharyngeal- or laryngopharyngeal tonsils (similar to pigs), or within the nasopharyngeal mucosa (similar to cattle) [25,29]. 

Despite the differences in the anatomic location of primary infection in different host species, the characteristics of the affected epithelial regions are strikingly similar. The epithelium that is specifically susceptible to primary FMDV infection has been referred to as reticular or lymphoid/follicle-associated epithelium (FAE) due to the consistent direct association with MALT [42,43,47]. In comparison to the surrounding, non-lymphoid epithelium, the FAE is thinner, with a discontinuous basal membrane, and abundant intra-epithelial leukocytes [48]. Shortly following primary epithelial infection, FMDV capsid proteins can be localized to antigen-presenting cells in the subjacent MALT [12,42]. However, replicating FMDV has only consistently been localized to cytokeratin-expressing epithelial cells [42,43,44]. The mechanisms of interaction of FMDV with the cells of the subepitehlial MALT during early infection and the manner of establishment of viremia remain to be elucidated. 

The viremic/clinical phase of disease is accompanied by development of characteristic vesicular lesions within and around the oral cavity, on the feet, and on udders [8,49]. Vesiculation includes massive amplification of FMDV within the keratinocytes of lesion sites [50,51]

FMDV induces a strong systemic immune response, and FMDV-specific antibodies can be detected in serum as early as 4–7 days after virus exposure [52,53,54,55]. Viremia is cleared shortly after the appearance of neutralizing antibodies, although infectious virus may remain viable in peripheral lesion sites for up to approximately 7–14 days [39,56]. In pigs, infectious FMDV is cleared from all tissues within 4 weeks of infection [34]. However, a substantial proportion of FMDV-infected domestic and wild ruminants remain infected within the same epithelial (FAE) regions of the upper respiratory tract that support primary infection for months to years after infection; a condition referred to as FMDV persistence or the carrier state [7,57,58], which is the subject of the remainder of this review. 

## 3. The FMDV Carrier State

### 3.1. Early Studies and Methods for Identifying FMDV Carriers

The carrier state of FMDV was first described by van Bekkum et al. in 1959. That early publication was based on accumulated field evidence and experimental data, and concluded that a substantial proportion of FMDV-infected cattle remained infected, as determined by virus isolation from “saliva” samples, up to several months after infection [59]. It was additionally reported that disease transmission from these carrier animals was deemed unlikely [59]. An optimized approach for the detection of persistently infected FMDV carriers by sampling of “oesophago-pharyngeal-”, or “oropharyngeal-“ fluid (OPF) by use of a probang cup was reported by Sutmoller and Gaggero [60]. However, this sampling approach was highly similar to the technique that had been used by van Bekkum in the preceding publication. OPF harvesting by probang sampling involves scraping of the mucosal surface of the oro- and nasopharynx with a metal cup attached to a metal rod. Several investigations have shown significant differences in FMDV detection in conventional swab samples compared to probang samples during the FMDV carrier state [39,40,41], suggesting that the contents included in probang-derived OPF samples are critical for the recovery of FMDV from persistently infected animals. This difference may be a consequence of the different anatomic sites of sample harvest, but is also likely affected by the cellular contents which are incorporated in the OPF sample due to the scraping action of the probang cup. 

Early experimental studies confirmed that FMDV exposure of cattle that had previously been subjected to active or passive immunization led to subclinical infection of the upper respiratory tract, and subsequent persistent infection, despite complete clinical protection [61]. Interestingly, that early publication by Sutmoller et al. also reported that the FMDV carrier state was not detected in immunized cattle that were challenged by intra-muscular injection of virus, further emphasizing the critical involvement of the bovine upper respiratory tract in establishment of primary and persistent FMDV infection. Subsequent years provided a number of publications confirming the occurrence of persistent FMDV infection in ruminants by isolation of FMDV from OPF samples from cattle, goats, sheep, and African buffalo [27,62,63]. A more recent publication demonstrated increased sensitivity of FMDV detection in persistently infected African buffalo in “tonsil swab” samples compared to OPF, when the tonsil swabs consisted of samples obtained by accessing the palatine tonsil sinuses of sedated animals using small nylon brushes [64].

Persistent FMDV infection in bovines has traditionally been defined in relation to an arbitrary threshold of 28 days, which was originally defined on the basis of experimental logistics, rather than biological data [61]. By this conventional standard, an animal from which infectious FMDV can be detected beyond 28 days after infection is considered an FMDV carrier [65]. However, more recent experimental studies have demonstrated that cattle that clear infection prior to becoming carriers generally do so substantially earlier than previously assumed [39,66]. Specifically, in one set of experiments, it was demonstrated that carrier status could be defined as early as 10 days in vaccinated cattle and 21 days in non-vaccinated cattle. The achievement of a more precisely defined timeline of the FMDV carrier state divergence enabled the definition of the transitional phase of infection, which represents the period during which clearance of infection occurs (Figure 1) [39,67].

### 3.2. Anatomic Localization of Persistent FMDV in Cattle

Based on the consistent recovery of FMDV in OPF (probang samples), but not in other sample types, the location of persistent FMDV infection has commonly been incorrectly referred to as the oropharynx. However, studies investigating the anatomic localization of persistent FMDV in tissues from cattle have repeatedly reported that detection is limited to tissues of the nasopharynx [57,68,69], specifically the dorsal surface of the soft palate and the adjoining dorsal nasopharynx [39,44,70] (Figure 2).

A detailed study published in 1966 concluded that the most consistent isolation of FMDV in post-mortem tissue samples from cattle occurred in the dorsal soft palate and pharynx [68]. The term “pharynx” was not further defined in that publication, but the description within the text, as well as distinction from other defined sample sites, suggest that it would likely have represented the nasopharynx. It is also noteworthy that the dorsal surface of the soft palate is the anatomical floor of the nasopharynx and shares a continuous mucosal surface with the dorsal nasopharynx (Figure 2). This early study by Burrows was based upon the isolation of FMDV from macerated whole tissue samples, and was not able to provide any resolution of whether the detected virus was derived from lymphoid or epithelial compartments of the tissues. Subsequent investigations specifically confirmed the localization of persistent FMDV to the epithelial surface of the dorsal soft palate by use of in situ hybridization [71,72]. Additionally, detection of FMDV RNA in samples from the dorsal surface of the soft palate was subsequently demonstrated to correlate with isolation of FMDV from OPF, providing further evidence of the importance of this specific anatomic location for FMDV persistence [56].

More recent works have expanded upon the knowledge of the anatomic location of persistent FMDV in cattle by demonstrating detection of structural- and non-structural FMDV antigen within epithelial cells of the bovine nasopharyngeal mucosa [39,44,70] (Figure 3). Specifically, simultaneous co-localization of FMDV VP1 and 3D proteins by immuno-microscopy was found in the same distinct segments of follicle-associated epithelium (FAE) directly overlying subepithelial MALT follicles wherein primary infection was identified [39]. Although infected cells were only detected in these FAE regions, only a minority of FAE regions were ever affected in any animal. Importantly, these studies also included broad sample collections of tissues from the oropharynx, lungs, lymph nodes, tonsils, as well as internal organs and distant anatomic sites, which were uniformly demonstrated to not contain FMDV during the carrier phase. Additionally, the sites of FMDV persistence were found to be identical in vaccinated cattle that had been fully protected against clinical FMD. Thus, the site of FMDV replication during persistent infection in cattle was confirmed to be identical to the site of primary infection, regardless of the occurrence or severity of clinical FMD. The same investigations also demonstrated concurrent detection of FMDV RNA, as well as rare detection of FMDV capsid protein, but not non-structural proteins, within the associated subepithelial MALT follicles [39,44]. Due to the lack of detection of non-structural proteins or isolation of infectious virus, this detection within lymphoid regions was concluded to represent viral degradation products rather than active viral replication. 

The localization of FMDV genome and structural protein to lymphoid tissues was emphasized in a publication from 2008, which demonstrated detection of FMDV capsid protein within germinal centers of lymph nodes draining the oral cavity and pharynx [73]. It was hypothesized that this detection represented the capture and retention of intact virions within the follicular dendritic cell (FDC) network, as had previously been described during HIV infection [74]. However, neither confirmation of viral replication occurring in these regions, nor demonstrated viability of potentially retained viral particles has ever been achieved. Overall, these findings support the involvement of FDCs and lymph nodes in the enduring immune response, but not as a site of viral replication during FMDV persistence. Thus, similar to early stages of FMDV infection in cattle, the specific mechanisms of interaction, or potential viral translocation, between the epithelium and MALT have not been fully elucidated.

### 3.3. Anatomic Localization of Persistent FMDV in Asian and African Buffalo

Studies of experimental FMDV infection in African buffalo (*Syncerus caffer*) demonstrated that the prevalence of recovery of FMDV from persistently infected animals was greater in palatine tonsil swabs compared to probang samples [64]. These tonsil swabs were collected using nylon brushes inserted into the tonsillar sinuses of sedated animals. Additionally, the recovery of FMDV from postmortem tissue samples from the same animals suggested that the pharyngeal- and palatine tonsils and nasopharyngeal mucosa (dorsal soft palate) are the anatomic sites of FMDV persistence in African buffalo [64]. These findings emphasize that despite similarities in susceptibility to infection, there are important differences in FMDV pathogenesis across distinct host species. 

There are no published descriptions of the tissue distribution of persistent FMDV in Asian buffalo (*Bubalus bubalis)*. However, the recovery of FMDV from tissues collected at slaughter points from clinically normal Asian buffalo indicates localization of persistent infection to the nasopharyngeal mucosa [75]. Further experimental studies have confirmed that the infection dynamics and systemic host response to FMDV infection in Asian buffalo, including the prevalence of FMDV persistence, are comparable to cattle [20,21,76]. 

### 3.4. FMDV Persistence in Small Ruminants

The FMDV carrier state in small ruminants has received little attention in comparison to other host species. An early experimental investigation by Burrows confirmed that the prevalence of persistent infection in sheep at 4 weeks post infection was comparable to earlier studies in cattle, but decreased substantially within 4–5 months after exposure [27]. The same investigation concluded that FMDV could be recovered from pharyngeal tissues of persistently infected sheep, but that in contrast to cattle, virus recovery was more likely from tonsils than mucosal samples [27]. Similar findings were subsequently confirmed, indicating that high proportions of both sheep and goats became FMDV carriers after contact exposure or intranasal instillation and that nearly 50% of sheep remained as carriers 9 months after infection [77]. A recent publication confirmed the localization of persistent FMDV to epithelial crypts within ovine oropharyngeal and laryngopharyngeal tonsils [29]. This finding was similar across naïve and vaccinated cohorts of sheep. However, the administration of a high payload vaccine (> 6 PD_50_) 14 days prior to virus challenge prevented FMDV persistence, even though all animals were confirmed to have been infected after challenge [29,78]. The combined output of FMDV pathogenesis studies in sheep thus suggest that although virus detection during early infection shares similarities with both cattle and pigs, the anatomic localization of persistent FMDV may be more similar to infection in African buffalo. Detailed characterization of persistent FMDV in goats is lacking.

### 3.5. Anatomic and Physiologic Considerations of Detection of FMDV in Probang Samples

The demonstrated localization of persistent FMDV infection to the bovine nasopharynx, despite consistent recovery of virus in sampled oropharyngeal fluid, has been a cause of some confusion, which can be readily clarified by consideration of the relevant anatomic structures (Figure 4). In cattle, persistent FMDV has consistently been localized to the mucosal surface of the dorsal nasopharynx and the adjoining dorsal soft palate (also part of the nasopharynx). These sites represent a continuous surface that makes up the tubular structure of the upper respiratory tract that extends from the nasopharyngeal tonsil (cranial aspect) to the larynx and the openings of the esophagus and trachea (caudal aspect) (Figure 2). When in a “neutral” breathing position, the caudal edge of the soft palate rests underneath the tip of the epiglottis to maintain an open path for air passing through the upper respiratory tract, from the nares, nasal cavity, and nasopharynx, down into the trachea and lungs (Figure 4A). However, when the animal swallows, this respiratory pathway is interrupted as the soft palate is lifted towards the dorsal nasopharynx, and the epiglottis closes off the tracheal entrance so that feed can pass from the oral cavity into the opening of the esophagus located above the larynx (Figure 4B). 

The insertion of a probang cup through the oral cavity of a cow induces a swallowing reflex. As the probang is passed through to the opening of the esophagus, the collection cup accesses and scrapes the nasopharyngeal mucosa, including the caudal aspects of the dorsal nasopharynx, as well as the caudal end of the dorsal soft palate as the cup is retracted (Figure 4C). Probang sampling is a semi-invasive procedure as it involves scraping of the mucosal surface with the edges of the metal cup. Postmortem examinations performed shortly after the collection of probang samples often reveal visible bruising at the caudal aspect of the dorsal soft palate, further demonstrating the harvesting of material from this region. FMDV cannot be detected in regular oral or nasal swab samples collected from cattle during persistent infection [39,40,41], further suggesting that there is minimal “natural” shedding of virus into the environment associated with FMDV carriers. 

Low quantities of infectious FMDV have been shown to be present in unadulterated OPF [79] which has been demonstrated to be infectious to naïve hosts upon direct transfer [80]. However, the infectivity of OPF harvested from carriers increases following treatment with fluorocarbon compounds such as 1,1,2-trichloro-1,2,2-trifluoroethane (TTE), which is believed to mediate the release of antibody-complexed virus [79,81]. This has been suggested to be related to opsonization of FMDV in OPF by secreted IgA. However, there has been no direct evidence of the presence of antibody-complexed FMDV in OPF from carriers, and the demonstrated increase in infectivity associated with freon or TTE treatment of OPF samples could potentially also be explained by the release of FMDV from intact cells, or membrane-covered cellular compartments. Further investigations are needed to confirm associations between FMDV and cellular components or secreted antibody in OPF, as well as the potential mechanisms of non-lytic release of virus from infected cells.

## 4. Host Responses as Related to the Carrier Phase

Cumulative evidence suggests that the mechanistic determinants of the FMDV carrier state are factors related to hosts’ immunological responses rather than intrinsic viral factors. There has been substantial progress in recent years in elucidating these mechanisms via various experimental approaches, suggesting determinants that fall into the general categories of innate factors, humoral immunity, and cell-mediated immunity.

### 4.1. Acute Phase Proteins and Early Anti-Viral Response

Acute FMD infection in cattle is associated with a significant but transient antiviral response that is quantifiable by a systemic peak in type I/III interferon (IFN) activity that coincides with the onset of viremia [52,53,82,83]. Additionally, early FMD infection has been shown to generate substantial peaks in serum levels of acute phase proteins haptoglobin (HP) and serum amyloid A (SAA) [53]. In contrast to findings from other chronic diseases of cattle [84], SAA and HP levels returned to near baseline levels during the FMDV carrier state [53]. However, total serum quantities of HP were found to be higher in cattle that successfully cleared FMDV infection compared to FMDV carriers [53]. As HP has been directly associated with inhibition of Th2 cytokine release [85,86], this finding may suggest a potential difference in the Th1/Th2 balance in carriers versus non-carriers, which would be consistent with findings from subsequent studies [87]. 

### 4.2. Humoral Response

FMDV-infected cattle develop a strong anti-FMDV serological response that can be detected as early as 4-7 days after infection [55,82,88]. Early FMDV infection is associated with a transient peak in anti-FMDV IgM followed by a sustained IgG response [52,55] Overall, this response is similar in cattle that efficiently clear infection, and those that maintain persistent infection, suggesting that the serological response is not a defining factor in establishment of the carrier state [52,53,55]. Additionally, in contrast to some persistent viral infections of humans, such as hepatitis B and C, there is no prolonged IgM response associated with FMDV persistence [55].

In contrast to the similarities of IgG and IgM responses across animal cohorts, multiple studies have reported significant differences in the anti-FMDV IgA response between carriers and non-carriers. Specifically, the FMDV carrier state has been associated with a prolonged IgA response in oral and nasal secretions [39,89,90] as well as in serum [55]. This suggests a sustained stimulation of a mucosal immune response that is directly associated with FMDV persistence. The sustained detection of anti-FMDV IgA in saliva may be utilized for the diagnostic identification of carriers and has implications regarding the immunopathogenesis of persistent infection in the nasopharynx. Furthermore, the presence of secreted IgA in oral secretions suggests that any non-cell associated FMDV shed during late stages of infection would likely be neutralized through opsonization, thereby further reducing the potential of FMDV transmission from carriers. 

### 4.3. Duration of Immunity

There is limited information regarding the duration of immunity following FMDV infection. One study by Cunliffe reported that convalescent cattle were clinically protected against re-challenge with homologous and heterologous strains of FMDV serotype O at 11 months after initial infection [91]. Additionally, one out of three steers that were challenged at 5 years post initial infection was also protected from clinical FMD. However, that study did not include any reports of virus recovery from convalescent animals, and it is therefore unclear if the cattle were persistently infected or not. More recent studies have suggested that under natural conditions, carriers may become superinfected with heterologous or homologous virus strains [38].

### 4.4. Cellular Response

The cell-mediated immune response to FMDV-infection has received substantially less attention compared to the serological response. Furthermore, most studies that have characterized the cellular response to FMDV infection have focused on the early stages of infection [82,92], and a majority of investigations have been carried out in pigs [93,94,95]. One experimental study demonstrated that in cattle, depletion of CD4+ T cells had no effect on the severity of clinical disease, viremia, or neutralizing antibody titers following FMDV challenge of naïve cattle [88]. This suggests that the rapid neutralizing response to FMDV infection is T cell independent, which has been further corroborated by the demonstration of detection of antibody-secreting cells in lymph nodes associated with the respiratory tract during very early stages of FMDV infection in cattle [96].

There are documented differences in the innate immune response to FMDV between cattle and pigs [97]. Nonetheless, there is a remarkable gap in knowledge of how species-specific differences in the host’s response to infection correlate with the demonstrated different outcomes of infection. Specifically, it is unknown if potential differences in the cell-meditated immune response of cattle and pigs may be responsible for persistence in cattle versus complete clearance of FMDV infection by pigs. One publication provided a detailed overview of the systemic immune response to FMDV in naïve and vaccinated cattle through both early and late stages of infection [52]. Although that investigation demonstrated a relative lymphopenia during acute FMDV infection of non-vaccinated cattle, there were no discernible differences in leukocyte counts, nor in CD4^+^ or CD8^+^ absolute counts or proportions that correlated with clearance versus persistence of FMDV [52]. A subsequent publication demonstrated significantly greater quantities of CD3^+^ and CD8^+^ lymphocytes in close proximity to segments of follicle-associated epithelium within the nasopharyngeal mucosa of cattle that had recently cleared infection, compared to those that were persistently infected [87]. Although an antibody-mediated response combined with phagocytosis is believed to be largely responsible for the clearance of viremia, direct cell-mediated immunity is required for the clearance of intra-cellular virus. Thus, further elucidation of the cellular host response in FMDV-infected tissues at transitional and persistent phases of infection could likely provide further insights into the mechanisms of virus persistence. 

### 4.5. Transcriptomics

#### 4.5.1. *In* *Vivo*

Alterations in gene expression patterns in tissues persistently infected with FMDV have been explored using qRT-PCR-, microarray-, and NGS-based platforms. Early studies that quantitated the expression of mRNAs by qRT-PCR demonstrated that genes with known anti-viral or pro-inflammatory functions were generally down-regulated in association with FMDV persistence [39,70]. One investigation reported significant negative correlations between detected quantities of FMDV RNA and IFN-α, -λ, CXCL10, and IRF-7 mRNA in samples of micro-dissected nasopharyngeal epithelium of FMDV carriers [39]. By contrast, two distinct investigations have reported significantly higher expression levels of TNF-α in nasopharyngeal tissues from FMDV carriers compared to non-carriers [98,99].

Subsequent studies utilized a bovine whole genome microarray platform to compare gene expression in whole-tissue macerates of nasopharyngeal tissue samples of carriers and non-carriers [100]. One study reported significant differences in gene expression patterns, suggesting that Th2 polarization, induction of immunologic tolerance, and inhibition of apoptosis were associated with the FMDV carrier state [100]. A follow-up investigation including samples from an overlapping set of study animals used the same microarray platform to investigate gene expression in samples of micro-dissected follicle-associated nasopharyngeal epithelium obtained from carriers and non-carriers, during both transitional and persistent phases of infection [87]. The latter investigation expanded upon the previous study by demonstrating that the Th2 polarization and inhibited apoptosis during the FMDV carrier state were specifically localized to the epithelial tissue compartment. Additionally, that report suggested up-regulation of cell-mediated immunity and Th1-associated pathways during the transitional phase of infection in animals that had recently cleared infection. A common factor for all of the mentioned investigations is that they have been based on limited sample sizes and have utilized one single assay platform without corroborating analyses. Thus, further studies involving larger sample sizes and additional experimental approaches are required to confirm the presented interpretations. However, the unifying themes of transcriptomic analysis suggest that the carrier state may be associated with local upregulation of anti-apoptotic pathways in infected epithelium and the inhibition of Th1-associated cellular immunity. 

#### 4.5.2. *In* *Vitro*

Other investigations have studied gene expression profiles in vitro using persistently infected cultured cells, either derived from FMDV-susceptible cell lines [101,102,103] or the primary culture of bovine pharyngeal epithelial cells [104,105]. The in vitro approach is particularly attractive as it eliminates the ethical concerns of animal experimentation, is less resource demanding and may therefore overcome limitations associated with small group sizes and high within-group variance. However, such approaches must be interpreted with critical consideration of the extent to which ex vivo systems may accurately simulate processes in biologically complex organisms. In particular, these studies occur in the absence of the host’s immune system, which is likely the most critical component defining the maintenance and clearance of persistent infection.

In a study by O’Donnell et al., primary cell cultures were established from tissues harvested from the bovine nasopharynx [105]. The cultured cells were characterized through multiple passages and were subsequently used to generate primary cell cultures persistently infected with FMDV. Low-passage cultures contained a relative abundance of cytokeratin-expressing cells, indicating epithelial histogenesis. However, repeated passage of the uninfected cells led to alterations in phenotypic composition of the cultures, and a relative majority (> 80%) of passage 15 cells expressed vimentin, which signifies either the selection of cells of mesenchymal origin or mesenchymization of the majority of the cells. Persistently infected cultures were established by 23 passages of cells surviving after FMDV inoculation. At that time, approximately 10% of cells contained FMDV capsid antigen, and in contrast to findings from in vivo studies, none of the persistently infected cultured cells were epithelial (cytokeratin-expressing) cells [105]. In addition to alterations in the cultured primary cells, the study reported of changes in viral phenotypic characteristics, as suggested by altered plaque morphology, receptor utilization, and the ability to grow in different cell lines [105]. Overall, these findings demonstrate that in this ex vivo system, changes occurred in both the viral and host characteristics over time.

A more recent investigation of persistent FMDV infection in vitro was based on an ex vivo model of multilayered cells derived from the bovine dorsal soft palate [104]. The harvested cells were passaged only five times and were subsequently cultured in an air–liquid interface using PTEE membranes. The response to FMDV infection was assessed at early (24h) and late (28d) time points. The results suggest a significantly upregulated anti-viral response in infected versus control cultures, with a less pronounced activation during late infection [104]. Additionally, there was evidence of a down-regulation of apoptotic pathways in association with FMDV persistence, which is similar to findings from investigations based on gene expression analysis in nasopharyngeal tissue samples [87,100]. Similar to previous studies, the ex vivo cultured epithelial cells utilized by Pfaff et al. underwent a gradual process of mesenchymization, signified by a transition from cytokeratin to vimentin expression [106]. 

Although in vitro studies can contribute important knowledge regarding alterations in gene regulation in FMDV-infected cells, the most critical limitation of this approach consists of the complete lack of the cell-mediated and systemic immune responses. Thus, it is possible to characterize the response to infection on the level of the infected cells, but it may not be appropriate to extrapolate these findings to explain tissue-level host responses in live animals. 

## 5. FMDV Genomics during Persistent Infection

A limited number of studies have investigated changes in the FMDV genome in association with the establishment or maintenance of persistent infection. The most common conclusion of these studies have been that no specific mutations in the viral genome were consistently associated with persistent infection [41,67,107,108,109]. Two separate investigations identified two different amino acid substitutions, VP1 Q172R and VP2 Y80H, which were consistently found in viruses recovered from persistent phase OPF samples within each of the investigations [110,111]; however, these findings were not corroborated by further studies. The intra-epithelial location of replicating FMDV during persistent infection can be presumed to be, to some extent, immuno-privileged, effectively reducing selective pressures on the virus. This is consistent with experimental findings confirming that there were no differences in the capacity of serum obtained from persistently infected African buffalo between 14 and 400 days post infection to neutralize virus obtained from the same animals at different times post infection [112]. That finding suggests that there is no pressure on the persisting virus to escape neutralization by serum-derived antibodies. 

Recent studies suggest that the rate of accrual of genomic changes differs significantly during early stages of infection in clinically susceptible animals, compared to subclinical neoteric and persistent stages of infection [67]. Additionally, while acute infection was associated with pronounced stochastic variability in viral genomes, the virus populations stabilized during persistent infection [67]. However, while genomic variations observed during early infection reflected inherent variability present in the experimental inoculum, the accrual of novel mutations (not present in the infecting virus population) increased through later stages of infection [113]. Field studies have confirmed that changes to the FMDV genome, potentially affecting viral antigenicity, continue to occur during persistent infection [109,114]. There is, however, no evidence of such processes leading to the emergence of novel viral lineages [112]. As the greater extent of viral replication during acute infection coincides with a substantially higher likelihood of transmission between animals, viral genomic changes occurring during acute infection are likely of greater epidemiological significance. Overall, there are still limited data available to adequately characterize the contributory effects of viral genomic changes on the establishment and maintenance of the FMDV carrier state; however, ongoing innovations in next-generation sequencing and improved bioinformatics tools are likely to provide further clarity in the coming years. 

## 6. Epidemiological Aspects of the FMDV Carrier State

It is well documented that FMD outbreaks affecting domestic ruminants in endemic regions (i.e., when infected animals are not culled) will generate varying proportions of FMDV carriers within the affected population [59,115,116]. Experimental studies have suggested that carrier prevalence in cattle may commonly exceed 50% of infected animals at 4-6 weeks after infection [39,53,61,117,118]. However, determination of the FMDV carrier prevalence in field studies is more complicated, as sampled animals generally represent only a small fraction of the exposed population and the timing of virus exposure(s) is often unknown. Additionally, as virus recovery from OPF samples is highly sensitive to handling and storage conditions, FMDV detection in field samples is generally less consistent compared to experimental samples. A targeted surveillance study in Vietnam reported an overall carrier prevalence of 2.4% amongst cattle and Asian buffalo, as determined by the detection of FMDV in OPF [76]. However, the timing of the most recent virus exposure (outbreak) varied across targeted study areas, ranging from 2 months to 2 years. Using a different approach, Bronsvoort et al. reported a herd-level FMDV detection of 19.5%, and an animal-level prevalence of 3.4% from a cross-sectional study conducted in central Cameroon [119]. That study did not include information regarding the estimated timing of recent outbreaks within the study area, and therefore precludes the categorization of animals into subcategories of subclinical disease stages. An earlier study by Hedger reported a carrier prevalence of approximately 20% in domestic cattle in Botswana, at approximately 7-12 months after the most recent outbreak [120]. It is thus clear that the prevalence of FMDV persistence reported from field studies varies depending on factors such as study design, definition of animal categories, and sample management.

### 6.1. Duration of the FMDV Carrier State

Animals that are persistently infected with FMDV will eventually clear the infection. However, there is substantial variation between different studies reporting the rate at which this clearance occurs. For cattle, the duration of the FMDV carrier state is often cited as “up to 2 years”. This conventional wisdom can be traced back to an original publication reporting FMDV detection in one out of 66 monitored cattle, at 24 months after an outbreak of FMDV serotype C [121]. Less commonly cited is that the same publication also reported that half of that same cohort of study animals had cleared infection at 4 months after the outbreak [121]. Tenzin et al. [122] conducted a meta-analysis of published experimental studies to conclude a decrease in the proportion of detected carriers of 0.115/month. Other investigations have been based on longitudinal sampling of animals identified as FMDV carriers following naturally occurring outbreaks. A study published by Hayer et al. [115] was based on monthly sample collection of FMDV carriers from 6 to 23 months after a confirmed outbreak caused by FMDV strain O/ME-SA/Ind2001d at two Indian dairy farms. That study concluded that the average time to termination of the FMDV carrier state was 13 months, which defined a decrease in the proportion of carriers of 0.07 per month. A different field investigation following an outbreak of FMDV O/ME-SA/PanAsia in Vietnam concluded that the mean duration of the FMDV carrier state in 10 longitudinally sampled cattle was 27.7 months, with a monthly decrease in the proportion of carriers of 0.03% [109]. However, in the latter investigation, four of the 10 persistently infected cattle were still FMDV-positive in OPF at the study conclusion 32 months after the outbreak [109]. In contrast to these longitudinal investigations, another study used a mathematical modeling approach based on cross-sectional field data to conclude that the probability of detecting an FMDV carrier at 12 months after an outbreak was less than 0.7% [123]. The approach by Bronsvoort et al. differs from the previously cited works, in that rather than calculating the rate of clearance of infection in a cohort of persistently infected animals, the model estimates the likelihood of detecting a carrier within a defined population, regardless of the previous infection status of those animals. A similarly modeled meta-analysis of longitudinal studies in India and Vietnam estimated 32%–51% probability of persistent infection 12 months post-outbreak among seropositive animals or previously identified carriers [124].

Overall, the integrative interpretation of these studies suggests that the duration of the carrier state is variable when examined in different contexts, and by different analytic approaches. It is likely that different reported durations are dependent upon the intrinsic attributes of specific viruses, host genetics, and sample handling across studies; however, it is difficult to discern the extent to which individual factors contribute to discordance. In summary, it seems clear that in a population of carrier cattle the prevalence of carriers decreases over time at a rate ranging from 0.03 to 0.1 per month, and that the herd level duration of the carrier state in cattle is likely between 6 and 24 months, but may be greater than 32 months in some contexts.

### 6.2. Transmission from FMDV Carriers

The risk of disease transmission from persistently infected FMDV carriers is a highly controversial topic. Neither re-activation of clinical disease nor vertical transmission has ever been documented from FMDV carrier cattle. The general consensus across studies is that FMDV transmission from persistently infected cattle is unlikely, whereas transmission from persistently infected African buffalo may be more common. Despite this consensus, the risk of contagion associated with subclinical FMDV carriers is often cited as the reason behind the extended waiting period that is required for a country to regain official status as free from FMD when vaccination has been used to control FMD outbreaks [125].

Despite several attempts, there are no published studies that have been able to confirm transmission from persistently infected cattle to in-contact sentinels [41,109,117]. Furthermore, immuno-suppression of experimentally infected FMDV carriers did not provoke transmission to in-contact sentinels or the re-occurrence of clinical disease in infected hosts [117]. A meta-analysis by Tenzin et al. [122] included mention of one published record of transmission from persistently infected cattle to contact-exposed pigs. However, the original study from which the observation was cited only reported confirmed seroconversion in two out of six pigs, which had been housed together with three persistently infected cattle for 75 days [126]. Without discrediting the findings of the original report, it is worth noting that the cited work involved long-term housing of susceptible animals within an experimental facility in which work with infectious FMDV was carried out, at a time when biosecurity practices were less stringent than modern day standards. 

In contrast to the majority of experimental and field data suggesting lack of contagiousness, one recent investigation confirmed that untreated OPF harvested from FMDV carriers caused fulminant FMD when deposited into the nasopharynx of naïve recipient cattle [80]. Although not a demonstration of natural disease transmission, the investigation did confirm that the FMDV that is present in the OPF of carrier cattle is infectious to susceptible animals. Additionally, as the challenge of the recipient animals did not involve needle-inoculation, the virus that was present in the carriers’ OPF was capable of overcoming the mucosal barrier and entering into susceptible epithelial cells within the nasopharynx of the recipient cattle. In contrast to the successful transfer of infection to cattle, the same study concluded that pigs were not infected by oropharyngeal deposition of the same material at similar doses. Additionally, feeding pigs with macerated nasopharyngeal tissues from the same cohort of persistently infected cattle did not result in infection in of any of the pigs [80].

Although the amount of infectious virus shed by persistently infected cattle is very low, the evidence confirming that this virus would indeed be capable of causing disease in susceptible animals confirms that there is detectable contagion associated with carriers and contributes to the uncertainty of the relevance of carriers at the individual animal level. Thus, even though the risk of a contagion associated with individual FMDV carrier cattle is likely minimal, the high numbers of carriers that are present following FMD outbreaks in endemic regions may still constitute a potential risk of virus dissemination, when considered at the population level. 

A limited number of published studies have demonstrated FMDV transmission from persistently infected African buffalo (*Syncerus caffer*) to sentinel cattle [127,128], although a more recent attempt to replicate the earlier findings concluded that transmission did not occur, despite up to 365 days of direct contact exposure [64]. It is believed that FMDV infection in herds of African buffalo is maintained, in part, through transmission from subclinically infected dams to neonates; however, the specific mechanism of transmission and the herd-level persistence of infection have never been determined or tested. Investigations based on phylogenetic analysis of viruses obtained from cattle and buffalo in FMD endemic regions have shown that FMDV populations found in the different host species are generally distinct, although some overlap suggests that inter-species transmission events do occur [129]. However, as FMDV infection in African buffalo is usually subclinical (neoteric), confirmed transmission of FMDV from buffalo to cattle does not necessarily imply the involvement of FMDV carriers. Similarly, a recent investigation has shown that substantial subclinical circulation of FMDV, including the introduction of new virus strains, can occur in herds of vaccinated Asian buffalo (*Bubalus bubalis*), in the absence of any signs of clinical FMD [38]. That investigation was based on a 12 month study with repeated OPF sampling of buffalo at 30 different dairy farms in Pakistan, and demonstrated that multiple introductions of new serotypes and lineages occurred on all farms, without any clinical cases of FMD [38]. Those recent findings are consistent with previous studies that have demonstrated a high prevalence of FMDV detection in oral swabs from Asian buffalo and cattle in endemic regions, without concurrent clinical evidence of FMD in the herds [130,131]. Experimental studies of FMDV infection in previously naïve Asian buffalo reported that although characteristic vesicular lesions were common, the buffalo were less likely to develop lameness or inappetence when compared to cattle [20]. This blunted manifestation of clinical FMD could potentially facilitate the dissemination of infection, as lesions are likely to go unnoticed in the absence of lameness, and clinically unaffected animals are more likely to be moved.

### 6.3. Epidemiological Concerns of Neoteric Versus Persistent Infection

The term “subclinical infection” is often used to specifically refer to the early stages of FMDV infection that occurs prior to, or in the absence of, clinical signs of disease [132]. This is imprecise usage of terminology, as the strict meaning of a subclinical infection is an infection without obvious signs of disease, and this term thereby also accurately describes the persistent phase of FMDV infection. As neoteric (new or temporally acute) subclinical infection entails unapparent infection combined with virus shedding, such animals are likely of greater relevance for FMD dissemination compared to FMDV carriers. Nonetheless, in field scenarios, it may not be practically feasible to discern between neoteric and persistent subclinical infection, specifically in cases where there have been no reports of FMD outbreaks. Regardless of this distinction, it is clear that the sampling of animals without apparent clinical disease may provide useful information regarding which viruses are in circulation within distinct geographic regions. In addition to informing local vaccination programs and PCP-FMD, such information may also provide valuable insights into the mechanisms of dissemination of distinct FMDV lineages between distant geographic regions. 

### 6.4. FMDV Persistence in Wildlife

With the exception of African buffalo, wild ruminants and suids are believed to be of limited importance for the maintenance and spread of FMD in endemic regions [7,133]. Spillover of infection into susceptible wildlife populations during FMD outbreaks in non-endemic countries may complicate disease control efforts, but it is unlikely that such events would lead to the establishment of a true wildlife reservoir of infection [133]. The FMD outbreak that occurred in California, USA, in 1924 may represent a notable exception, as control efforts involved culling of approximately 22,000 mule deer in the Stanislaus national forest, with reports of clinical FMD observed in 10% of culled deer [134,135]. Feral pigs and wild boar are susceptible to clinical FMD [136,137], but are, similar to domestic pigs [34], not capable of maintaining persistent FMDV infection. Experimental studies have also demonstrated susceptibility to infection in white tailed deer *(Odocoileus virginianus)*, red deer *(Cervus elaphus elaphus)*, pronghorn antelope *(Antilocapra americana)*, elk *(Cervus elaphus nelsoni)*, and bison *(Bison bison)*, but without detection of persistent infection in any of those species [138,139,140,141]. Similarly, clinical FMD without evidence of virus persistence has been observed in impala (*Aepyceros melampus*) [142]

## 7. FMD Vaccines and FMDV Persistence

### 7.1. FMD Outbreak Control by Vaccination

Conventional FMD vaccines have contributed substantially to the successful eradication of FMD in Europe and the majority of South America [143,144,145], and continue to play a critical role in FMD control in endemic regions. However, the use of vaccination to control FMD outbreaks in previously free countries is still controversial due to the financial implications associated with the prolonged waiting period required to prove freedom from FMD. Specifically, the OIE terrestrial code dictates a mandatory waiting period of 6 months in order to prove freedom from FMD following an outbreak when vaccination without subsequent stamping out or slaughter of vaccinated animals is deployed, compared to 3 months when vaccination is not used [146]. This lengthened embargo period is attributed to the perceived risk of unidentified FMDV carriers within a vaccinated population. In addition to embargos imposed in association with FMD outbreaks, the potential risk of contagion associated with vaccinated animals has had a substantial impact on the regulation of trade with countries in which FMDV vaccination is regularly practiced [147]. It is thoroughly established that currently available FMD vaccines do not prevent subclinical or persistent FMDV infection [39,41,59,68]; however, in a field setting, vaccination will likely lead to a reduced occurrence of FMDV persistence, as the overall disease burden at the population level is reduced and fewer animals will be exposed to the virus. Nonetheless, as vaccinated and infected cattle will likely go unnoticed if clinical signs of FMD are absent, the movement of such animals, especially during early stages of infection, may contribute to disease spread. 

### 7.2. The Discrepancy between Clinical Protection and Protection against FMDV Persistence

For vaccines to be truly efficient in preventing subclinical and persistent FMDV infection, they must ultimately be designed to induce protection at the mucosal barrier that is sufficient to prevent primary infection; a concept that has been largely overlooked in FMD vaccine research. Specifically, FMD vaccine development has traditionally emphasized the capacity of a vaccine to induce a strong humoral immune response, with vaccine efficacy evaluated based upon the capacity to prevent disseminated infection (defined by lack of lesions distant from the virus inoculation site) [148]. More recently, studies focusing on the induction of neutralizing antibody titers post vaccination have been used as proxies for vaccine efficacy [149,150,151]. However, as repeatedly demonstrated in experimental studies; the presence of neutralizing antibodies in serum pre-challenge, or the lack of clinical lesions post challenge, are not reliable predictors of a lack of infection in vaccinated cattle [39,41,52,117,152]. Additionally, a majority of infected vaccinated cattle remain infected for a longer duration (carriers). 

FMDV infection induces a strong antibody-mediated immune response, which is believed to be critical for the efficient clearance of viremia. There is thus a strong association between a Th2-weighted host response and clinical recovery, which may explain the high prevalence of persistently infected cattle after natural infection. As a Th2-biased immune response is inefficient in clearing intracellular virus, stimulation of adequate cell-mediated immunity should be a goal for development of the next generation of FMD vaccines in order to attempt to mitigate the establishment of FMDV carriers. 

### 7.3. Current OIE-Guidelines for Vaccine Trials Do Not Consider Carrier Prevention

In addition to an exaggerated focus on neutralizing antibody titers, the OIE-mandated design of vaccine trials precludes conclusions regarding protection against subclinical neoteric and persistent infection. Specifically, virus challenge in vaccine efficacy trials in cattle is performed by injection of 10,000 infectious doses of FMDV into the tongue epithelium [148]. This standardized approach provides a stringent and consistent challenge that is entirely appropriate to evaluate protection against clinical FMD. However, by injecting the virus into the tongue epithelium, the natural pathway of virus entry via the mucosa of the upper respiratory tract is completely bypassed. By consequence, if vaccine-induced immunity is efficient in preventing viral dissemination from the injection site, as is often the case with high-payload vaccines and homologous challenge, the nasopharynx, which in cattle is the only anatomic site that is permissive to subclinical primary and persistent FMDV infection, will not be exposed to the virus. Thus, although tongue-inoculation will provide valid information regarding the vaccine’s capacity of protecting against clinical FMD, this system is not appropriate for the evaluation of protection against subclinical neoteric and persistent FMDV infection. 

## 8. Concluding Remarks

Since its’ first description in the 1950s, the FMDV carrier state has been a subject of controversy. Despite over half a century of dedicated research efforts, there are still remarkable knowledge gaps related to the underlying mechanisms of persistent FMDV infection. Despite these gaps, recent years have provided substantial progress in detailed characterization of the FMDV carrier state, specifically in relation to determining the micro-anatomic localization of replicating virus and elucidating host response mechanisms that may facilitate virus persistence. Cumulative evidence suggests that the factors determining the establishment and endurance of FMDV persistence are more likely related to host factors rather than viral determinants. Despite this, little effort has been invested into the development of improved vaccines that could effectively prevent the FMDV carrier state. Improved understanding of specific aspects of the host response that are associated with the FMDV carrier state divergence is needed to guide the development of the next generation of vaccines and biotherapeutics which may ultimately mitigate the FMDV carrier state problem. Detailed understanding of the natural mechanisms which make pigs resistant to persistent infection may provide a model which could be emulated to improve immune responses in carrier-susceptible species. A key component in such endeavors should be to focus on achieving a mucosal response sufficient to prevent the primary infection of vaccinated animals. For this to be a realistic goal, the standard experimental models and assays used to evaluate FMD vaccine efficacy will also need to be adapted to enable appropriate evaluation of vaccine-induced protection against subclinical neoteric and persistent infection.

In FMD-free countries, FMDV persistence is often cited as an important impediment for the willingness to use emergency vaccination to control potential FMDV incursions. With growing public concerns related to sustainability and environmental impacts of animal production, mass depopulation of livestock to control disease outbreaks is likely to become increasingly problematic. While there may be strong financial incentives to avoid vaccination in order to regain a status as FMD-free as soon as possible [153], the long-term economic impacts of alternate control options, including vaccination and monitoring of infected herds, should be evaluated in the context of large scale FMD outbreaks. Specifically, as seen within the past decade in South Korea, a strict approach of stamping out to maintain freedom from FMD without vaccination may not be feasible when the disease pressure is overwhelming [6,154,155,156,157,158]. Additionally, when indirect costs associated with FMD outbreaks, such as job losses within the livestock sector, and losses associated with reduced tourism are accounted for, the estimated financial benefits of vaccination often increase [159,160,161].

There is a general consensus that the risk of disease transmission from FMDV carrier cattle is minimal. This is a stark contrast to persistent infection with bovine viral diarrhea virus (BVDV) in cattle, in which persistently infected animals are highly important as reservoirs due to the shedding of high quantities of infectious virus [162]. However, given the substantial numbers of persistently FMDV-infected ruminants that are present in endemic regions, combined with extensive transboundary movements of these animals, this minimal animal-level risk may still translate to a legitimate population-level risk. Nonetheless, neoteric subclinical infection likely represents a substantially greater risk for FMDV dissemination compared to persistent infection. As these two concepts are similar with regards to the lack of apparent disease, the distinction of the two is challenging in relation to disease control in endemic regions. Thus, active surveillance of animals without apparent clinical signs of FMD coupled with molecular epidemiological analyses should be routinely applied to gain better understanding of transmission chains, region-specific endemicity, and long-distance movements and evolution of FMDV. 

Despite the consensus that transmission from FMDV carriers is exquisitely improbable, it is highly unlikely that the OIE and FMD-free states would ever tolerate the survival of carriers consistent with achieving FMD-free status. The carrier conundrum is therefore unlikely to be resolved until products are developed that can prevent or cure the FMDV carrier state.

## Figures and Tables

**Figure 1 pathogens-09-00167-f001:**
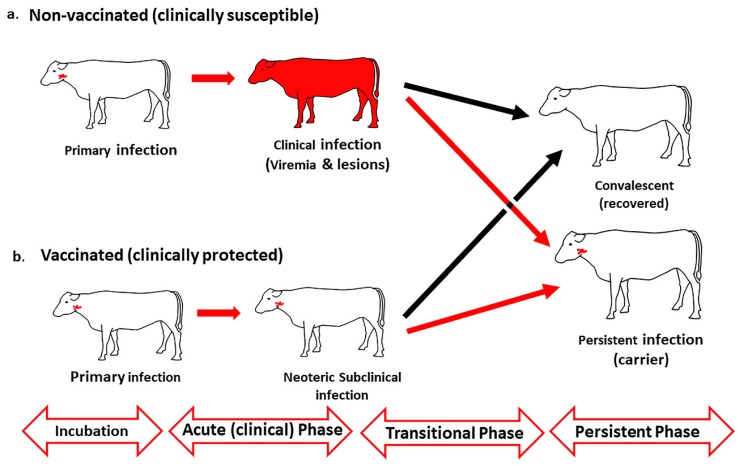
Temporal progression of foot-and-mouth disease (FMD) in naïve and vaccinated cattle. (**a**) In clinically susceptible (non-vaccinated) cattle, primary foot-and-mouth disease virus (FMDV) infection in the upper respiratory tract (nasopharynx) is followed by systemic generalization concurrent with viremia and development of vesicular lesions. (**b**) In vaccinated (clinically protected) cattle, FMDV infection remains restricted to the nasopharynx, and the primary phase of infection is followed by a phase of neoteric subclinical infection during which infected animals may shed infectious virus in oral- and nasal secretions. Both clinically susceptible and protected cattle traverse the transitional phase, during which animals will either clear infection or establish persistent infection (FMDV carriers).

**Figure 2 pathogens-09-00167-f002:**
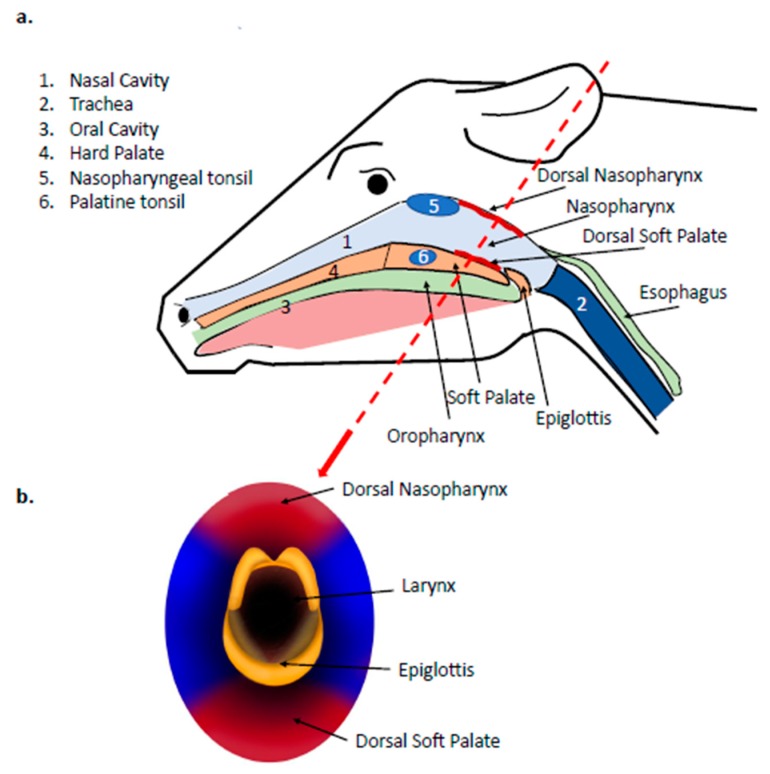
Anatomy of the bovine upper respiratory tract relevant to FMDV infection. (**a**) The bovine nasopharynx is the anatomic connection between the nasal cavity (1) and the trachea (2). The dorsal surface of the soft palate is the anatomical floor of the nasopharynx and shares a continuous mucosal surface with the dorsal nasopharynx (both surfaces emphasized in red). (**b**) A frontal plane view of the bovine upper respiratory tract at the level of the nasopharynx clarifies the direct continuation of the dorsal surface of the soft palate and dorsal ceiling of the nasopharynx, as these structures comprise a tubal compartment through which air passes en route to the larynx and tracheal opening.

**Figure 3 pathogens-09-00167-f003:**
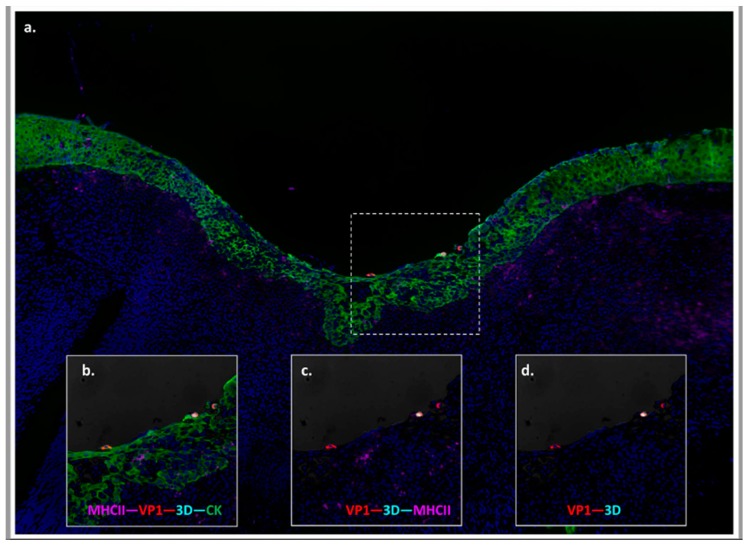
Localization of persistent FMDV to follicle-associated epithelium of the bovine nasopharyngeal mucosa. (**a**) FMDV infection in the epithelial surface of the dorsal soft palate at 35 days post contact exposure. FMDV VP1 (red) and 3D (teal) are localized to scattered cytokeratin^+^ epithelial cells (green) within a segment of follicle-associated epithelium overlying a subepithelial lymphoid follicle of MHC II^+^ cells (purple). Multi-channel immunofluorescence, 10x magnification (**b**–**d**) Magnification of region of interest showing merged and separate image channels—40x magnification.

**Figure 4 pathogens-09-00167-f004:**
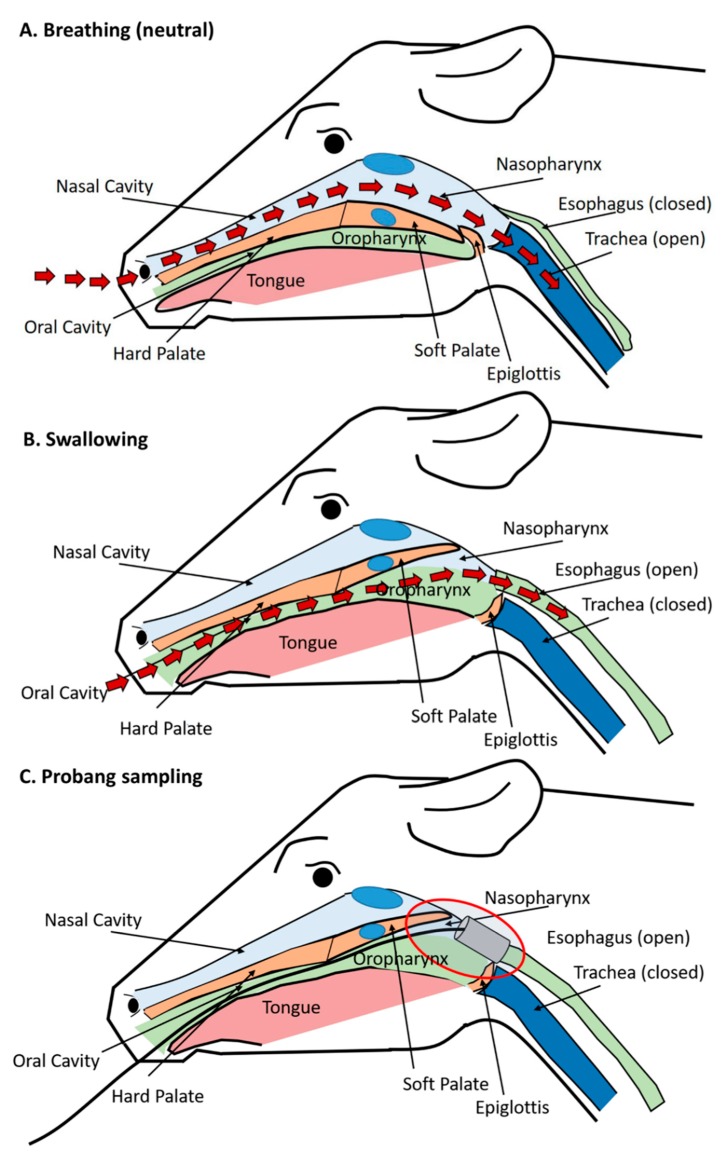
Anatomic and physiologic considerations of detection of FMDV in probang samples. (**A**) When the animal is breathing, the caudal edge of the soft palate is placed in a resting position underneath the tip of the epiglottis to maintain an open path for air passing from the nares and nasal cavity, through the nasopharynx, continuing into the trachea and lungs (path of red arrows). (**B**) A swallowing reflex is associated with realignment of structures so that the soft palate is elevated towards the dorsal nasopharynx, the epiglottis closes access to the trachea so that feed can pass from the oral cavity and oropharynx, passed the larynx and into the opening of the esophagus. (**C**) Insertion of a probang cup into the bovine oropharynx induces a swallowing reflex which allows access of the metal cup to the dorsal nasopharynx as well as the caudo-dorsal soft palate (this “capture” of the caudal end of the soft palate is often felt as a firm resistance upon retraction of the probang cup and can be evidenced by visible bruising upon postmortem examinations).

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
