# Peer review of "The Carrier Conundrum; A Review of Recent Advances and Persistent Gaps Regarding the Carrier State of Foot-and-Mouth Disease Virus"

_pathogens, 2020, doi:10.3390/pathogens9030167_

Round 1

Reviewer 1 Report

This paper reviews the carrier state for FMDV with a focus on implications for virus transmission. This review is thorough, well-written, and provides a useful synthesis of information on an economically important disease. The authors also thoughtfully connect the biology to implications for disease control. My comments are all minor and optional.

  1. The phrase 'neoteric infection' was new to me, and I recommend defining it at first use. This concept is defined later (L53-55, L84-91), but it would be helpful to briefly define it when it is first introduced.
  2. L62-63: “animals with natural or vaccine-mediated resistance progress through similar phases of infection with no clinically observable manifestation”. Can you define what you mean by resistance here? Does it occur through multiple mechanisms, or is it immune-mediated?
  3. I found Figure 1 to be very helpful, and I would recommend citing it earlier when the phases of infection are first discussed (section 2.1, in addition to where it is currently cited).
  4. It was not clear to me whether Figure 3 is original data presented in this paper, or whether this image is adapted from another source. This could be clarified in the paper.

Author Response

This paper reviews the carrier state for FMDV with a focus on implications for virus transmission. This review is thorough, well-written, and provides a useful synthesis of information on an economically important disease. The authors also thoughtfully connect the biology to implications for disease control. My comments are all minor and optional.

  1. The phrase 'neoteric infection' was new to me, and I recommend defining it at first use. This concept is defined later (L53-55, L84-91), but it would be helpful to briefly define it when it is first introduced.

Response: It is a good point. We have inserted (“temporally acute”) to briefly clarify  the concept of neoteric subclinical infection upon the first mention (line 36), but left the more through description for the later passages

  1. L62-63: “animals with natural or vaccine-mediated resistance progress through similar phases of infection with no clinically observable manifestation”. Can you define what you mean by resistance here? Does it occur through multiple mechanisms, or is it immune-mediated?

Response: We have clarified both the viral and host aspects of “resistance” (lines 72-73).

  1. I found Figure 1 to be very helpful, and I would recommend citing it earlier when the phases of infection are first discussed (section 2.1, in addition to where it is currently cited).

Response: An additional reference to figure 1 has now been inserted in passage 2.1 (line 76)

  1. It was not clear to me whether Figure 3 is original data presented in this paper, or whether this image is adapted from another source. This could be clarified in the paper.

Response: This figure was specifically generated for inclusion in this review. We are not aware of any specific mechanism to identify such, but will do so if prompted by editorial staff.

Reviewer 2 Report

The authors provide a review of persistent, subclinical FMD. Much remains unknown about these “carriers”, especially regarding how persistence is maintained, and the potential to transmit disease. These gaps must be filled in, in order to better understand and ameliorate risks from carriers. This review aims to identify what we already know, what we still need to know, and why we need to know it.

The introduction provided a good background on the issue and on the virus/disease. Overall, it is well written and easy to understand, even for readers outside the immediate field. The figures were also especially helpful.

Details on how the review was conducted should be added as available and appropriate (e.g. time range included in search, time frame when search conducted, keywords, any inclusion or exclusion criteria, etc…)?

There are a few topics/questions that could be considered for inclusion, though I understand the need for the review to remain focused and not cover everything under the sun.

-In the overview section 2 (or elsewhere if more appropriate), can anything additional be said about different serotypes/subtypes? E.g. does the overview and discussion of disease apply to all serotypes, or do some cause different severties of disease or disease with different progression? Is sublinical infection and carrier state more likely with some serotypes? Is infection in clinically protected (or naturally protected) more common with any?

-Have any other sites—besides pharynx and associated tissues—within the body (besides lymph nodes) been extensively tested to see if they also support the persistent infection? Perhaps this is not relevant with FMDV, but if other sites support infection during acute viremic phase, it would be interesting to know if they have been investigated for maintaining persistent infection as well.

-In the case of carrier animals, can the animal re-enter disease state? Super infection is touched on (which was a good addition), but can the persistent infection “re-activate” and cause disease? Can a carrier pass the virus to unborn fetus?

-Regarding how the virus maintains persistent infection, has anyone looked into whether defective interfering particles might be involved?

-Is FMDV unique in this regard, when it comes to persistence and carrier state (for common ruminant diseases)? I am more familiar with human examples, so this may be an irrelevant question for those in the field. I was wondering along the lines of, if most other common ruminant viruses don’t have this situation then perhaps it makes it even more interesting/important to study (but also harder). Whereas if there are a lot of other similar situations with other animal viruses, then perhaps it could be commented on as to what we can learn from those viruses (or why we can’t learn from them)? Again this may very well be outside the scope, but something potentially worth mentioning to enhance the significane if it is relevant.

Author Response

The authors provide a review of persistent, subclinical FMD. Much remains unknown about these “carriers”, especially regarding how persistence is maintained, and the potential to transmit disease. These gaps must be filled in, in order to better understand and ameliorate risks from carriers. This review aims to identify what we already know, what we still need to know, and why we need to know it.

The introduction provided a good background on the issue and on the virus/disease. Overall, it is well written and easy to understand, even for readers outside the immediate field. The figures were also especially helpful. 

Details on how the review was conducted should be added as available and appropriate (e.g. time range included in search, time frame when search conducted, keywords, any inclusion or exclusion criteria, etc…)?

Response:

We agree with the reviewer that documenting and reporting search strategies is extremely important for some types of review papers, particularly those reporting meta-analyses or summarizing voluminous bodies of work. In the current paper, we reviewed the relatively modest body of work available on FMD persistence in different species of livestock and wildlife that was available to the authors. We employed numerous searches in PubMed and Scopus during 2019-20 using the combinations of terms: foot-and-mouth disease, FMD, FMDV, persistence, pathogenesis, transmission, host response and vesicular disease. However, the specific searches that led to this body of work were not documented for the purpose of reporting within the paper. We do not feel that describing this informal strategy would add to the validity of the paper, but we will do so if the importance is emphasized by reviewers or editors.

There are a few topics/questions that could be considered for inclusion, though I understand the need for the review to remain focused and not cover everything under the sun.

-In the overview section 2 (or elsewhere if more appropriate), can anything additional be said about different serotypes/subtypes? E.g. does the overview and discussion of disease apply to all serotypes, or do some cause different severties of disease or disease with different progression? Is sublinical infection and carrier state more likely with some serotypes? Is infection in clinically protected (or naturally protected) more common with any?

Response: These are very valid points; however, most “understanding” of these subjects has come from conventional wisdom and anecdotal experience rather than rigorous testing. The clinical severity and host specificity may vary greatly across different strains of FMDV, although this is generally believed to be a strain-specific, rather than serotype-specific phenomenon. We have inserted a new passage in section 2.1 (lines 62-67) to further expand upon this concept.

-Have any other sites—besides pharynx and associated tissues—within the body (besides lymph nodes) been extensively tested to see if they also support the persistent infection? Perhaps this is not relevant with FMDV, but if other sites support infection during acute viremic phase, it would be interesting to know if they have been investigated for maintaining persistent infection as well.

Response: It is a relevant point, and we have provided additional clarification now, inserted in section 3.2 “Anatomical localization…”) (line 219). We have previously performed extensive investigations to elucidate potential sites of FMDV persistence in both cattle and pigs, but without detecting infectious virus anywhere other than the bovine nasopharynx.

-In the case of carrier animals, can the animal re-enter disease state? Super infection is touched on (which was a good addition), but can the persistent infection “re-activate” and cause disease? Can a carrier pass the virus to unborn fetus?

Response: To our knowledge, there is no evidence suggesting that persistent FMDV can be re-activated to cause clinical disease. An additional sentence to clarify this has been inserted in passage 6.2 (line 589-590). Persistent FMDV is continuously replicating within the nasopharyngeal mucosa. Maintained restriction to this anatomic region is likely mediated by the sustained high levels of serological immunity (which does not prevent super-infection by other virus types). There is one previous publication in which persistently infected cattle were immuno-suppressed in order to provoke transmission to in-contact animals, although with a negative outcome. Citation of this specific reference has been inserted (line 597-599).

Vertical transmission has never been documented from persistently infected carriers of FMDV; there is only minimal description of vertical transmission during the (acute) clinical phase in cattle and sheep. A sentence regarding presumed neonatal infection in African buffalo has been added in section 6.2 (line 634-636)

-Regarding how the virus maintains persistent infection, has anyone looked into whether defective interfering particles might be involved?

Response: We have searched the published literature, and have not found any investigations of this mechanism.

-Is FMDV unique in this regard, when it comes to persistence and carrier state (for common ruminant diseases)? I am more familiar with human examples, so this may be an irrelevant question for those in the field. I was wondering along the lines of, if most other common ruminant viruses don’t have this situation then perhaps it makes it even more interesting/important to study (but also harder). Whereas if there are a lot of other similar situations with other animal viruses, then perhaps it could be commented on as to what we can learn from those viruses (or why we can’t learn from them)? Again this may very well be outside the scope, but something potentially worth mentioning to enhance the significane if it is relevant

Response: This is a good point, but slightly beyond the scope of the current review. The most commonly recognized persistent viral infection in ruminants is Bovine Viral Diarrhea Virus (BVDV; pestivirus/Flaviviridae). The mechanisms exploited by BVDV are however, distinct from FMDV as persistent BVDV infection involves infection during the fetal stage, resulting in a subsequent lack of a serological response in the calf, which thereby becomes a persistently infected “super shedder”.